# Serological surveillance reveals a high exposure to SARS-CoV-2 and altered immune response among COVID-19 unvaccinated Cameroonian individuals

**Arlette Flore Moguem Soubgui**[1,2], **Wilfred Steve Ndeme Mboussi**[1,2], **Loick Pradel Kojom Foko** [2,3]*, **Elisée Libert Embolo Enyegue**[4], **Martin Luther Koanga Mogtomo**[1,2]*

**1** Faculty of Science, Department of Biochemistry, The University of Douala, Douala, Cameroon, **2** Centre de Recherche et d'Expertise en Biologie, Douala, Cameroon, **3** Department of Animal Biology, Faculty of Science, The University of Douala, Douala, Cameroon, **4** Center for Research on Health and Priority Diseases, Ministry of Scientific Research and Innovation, Yaoundé, Centre Region, Cameroon

* kojomloick@gmail.com (LPKF); koanga@yahoo.com (MLKM)

**Data Availability Statement:** All data are within the manuscript and supporting files, and relevant data

## Abstract

Surveillance of COVID-19/SARS-CoV-2 dynamics is crucial to understanding natural history and providing insights into the population's exposure risk and specific susceptibilities. This study investigated the seroprevalence of SARS-CoV-2 antibodies, its predictors, and immunological status among unvaccinated patients in Cameroon. A multicentre cross-sectional study was conducted between January and September 2022 in the town of Douala. Patients were consecutively recruited, and data of interest were collected using a questionnaire. Blood samples were collected to determine Immunoglobin titres (IgM and IgG), interferon gamma (IFN- γ) and interleukin-6 (IL-6) by ELISA, and CD4+ cells by flow cytometry. A total of 342 patients aged 41.5 ± 13.9 years were included. Most participants (75.8%) were asymptomatic. The overall crude prevalence of IgM and IgG was 49.1% and 88.9%, respectively. After adjustment, the seroprevalence values were 51% for IgM and 93% for IgM. Ageusia and anosmia have displayed the highest positive predictive values (90.9% and 82.4%) and specificity (98.9% and 98.3%). The predictors of IgM seropositivity were being diabetic (aOR = 0.23, $p$ = 0.01), frequently seeking healthcare (aOR = 1.97, $p$ = 0.03), and diagnosed with ageusia (aOR = 20.63, $p$ = 0.005), whereas those of IgG seropositivity included health facility (aOR = 0.15, $p$ = 0.01), age of 40–50 years (aOR = 8.78, $p$ = 0.01), married (aOR = 0.21, $p$ = 0.02), fever (aOR = 0.08, $p$ = 0.01), and ageusia (aOR = 0.08, $p$ = 0.01). CD4+, IFN-γ, and IL-6 were impaired in seropositive individuals, with a confounding role of socio-demographic factors or comorbidities. Although the WHO declared the end of COVID-19 as a public health emergency, the findings of this study indicate the need for continuous surveillance to adequately control the disease in Cameroon.

underlying the results presented in the study has been filed in the Dryad repository and available from https://doi.org/10.5061/dryad.1vhhmgr00.

**Funding:** The authors received no specific funding for this work.

**Competing interests:** The authors have declared that no competing interests exist.

## Introduction

A large number of pathogens have emerged and re-emerged in the last two decades and negatively impact the health, wellbeing, and economy of the world's populations. Several factors, greatly due to anthropic activities and climate change, have been indexed as the main causes of the appearance and/or resurgence of several infectious diseases [1]. The severe acute respiratory syndrome coronavirus 2 (SARS–CoV–2), a positive sense single-stranded ribonucleic acid virus, responsible for coronavirus disease 2019 (COVID-19) [2], is the latest example of this catastrophic scenario of emerging pathogens. Since its initial emergence in Wuhan, Hubei region, China, in December 2019, an estimated ~677 million cases and ~6.9 million deaths were attributed to SARS–CoV–2 as of 10 March 2023 (https://coronavirus.jhu.edu/map.html).

Methods relying on molecular detection of the viral genome, i.e., retrotranscriptase quantitative polymerase chain reaction (RT-qPCR), are gold standards for diagnostic and surveillance purposes. In developed areas, RT-qPCR is highly operational at the national level, but it is not the case in developing regions such as sub-Saharan Africa, where this technique is not affordable for most health facilities, especially those from remote and/or hard-to-reach population areas. In addition, RT-qPCR-based estimates do not reflect the real circulation and spread of SARS–CoV–2 in populations, as this technique can give false-negative results especially at initial testing (i.e., at the first healthcare encounter) [3], and is commonly recommended for individuals presenting COVID-19-like respiratory signs and symptoms [4]. In this context, it is crucial to develop alternative methods to overcome these main limitations of molecular techniques.

Serological testing for SARS–CoV–2 infection has become an important pillar in surveillance efforts. Seroprevalence studies quantify the number of individuals who have developed an immune response, i.e., antibodies, against a pathogen. The studies rely of the detection of immunoglobulins (e.g., IgG, IgM) produced followed infection with SARS-CoV-2. The production of these anti-SARS-CoV-2 antibodies corresponds to several phases of the natural course of viral infection. IgM are produced early in the humoral immune response against SARS-CoV-2 infection, and then they are switched to IgG, which has a higher affinity for viral antigens than IgM [5,6]. Not only can the large scale implementation of highly sensitive and specific serological tests help to know the real picture of spread of SARS–CoV–2 by tracking asymptomatic carriers of the virus who are missed by traditional surveillance systems, but these can also evaluate the effectiveness of control measures [5].

There is a lack of serological studies in African settings in the context of boosting efforts to continuous surveillance and control the disease. We therefore conducted a serological study to determine the prevalence and determinants of anti- SARS–CoV–2 antibodies among unvaccinated individuals living in the town of Douala, Cameroon. In addition, we evaluated variations in markers of immune responses (interferon, lymphocytes, and interleukins).

## Materials and methods

### Study design

This cross-sectional study was conducted between January, 1st and September, 31st, 2022 in seven health facilities in the town of Douala, Littoral Region, Cameroon. The study sites included Bangue district hospital, Boko health care centre, Bonassama district hospital, Cité des Palmiers district hospital, Deido district hospital, New Bell district hospital, and Nylon district hospital. These health facilities were selected as study sited based on three main reasons: ii) they were authorized by the government of Cameroon for diagnosis and treatment of COVID-19 patients, i) high frequency of patients' attendance, and iii) they were close to multifunctional reference laboratory for SARS-CoV-2 molecular diagnosis.

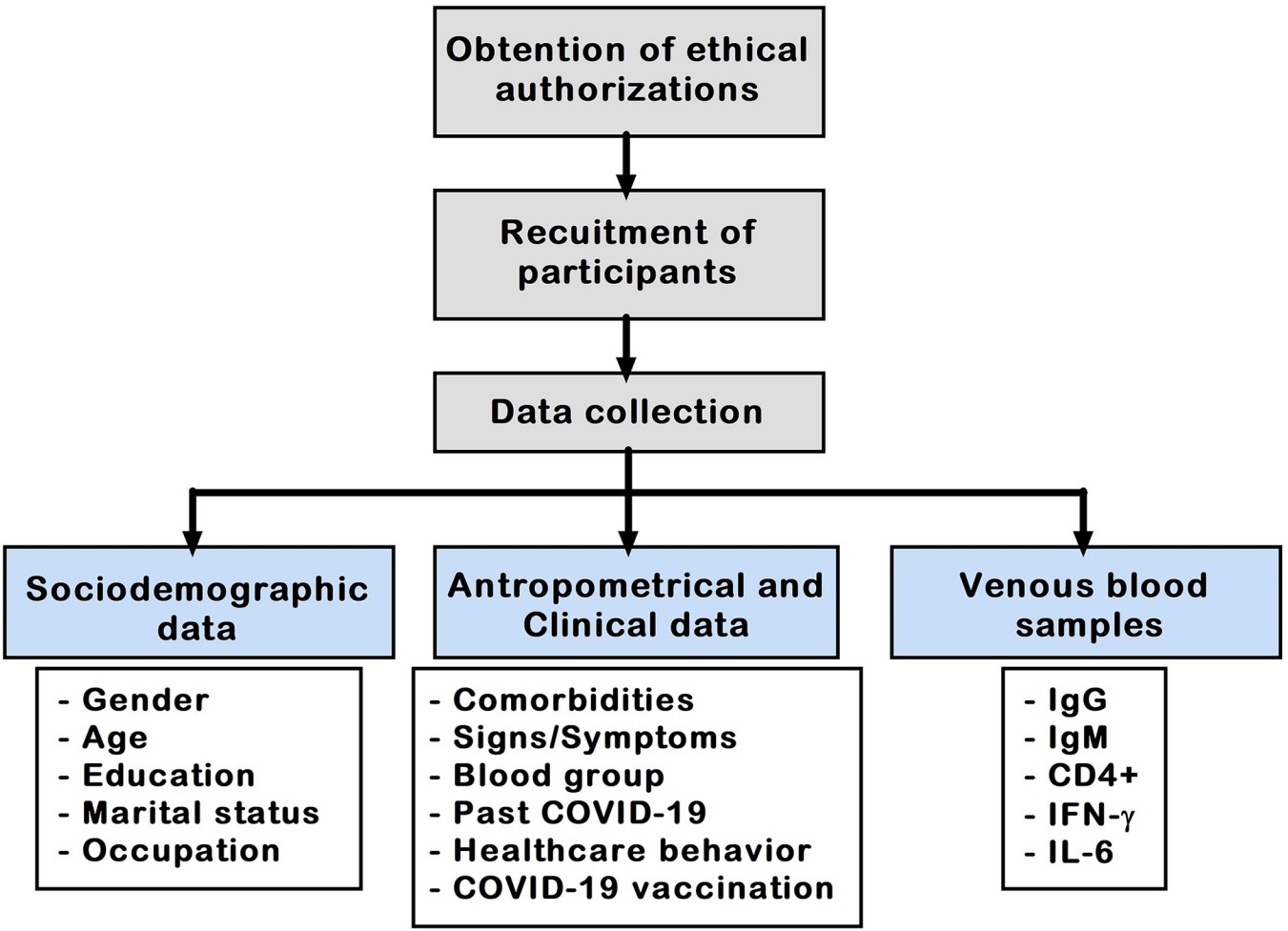

**Fig 1. Flow diagram of the study.** COVID-19: Coronavirus disease 2019, Ig: Immunoglobulin, CD: Cluster of differentiation, IFN-γ: Interferon gamma, IL-6: Interleukin 6.

Douala is the economic capital and most populated town of Cameroon. Populations living in this town are highly diverse, with the predominance of three ethnic groups (*Duala*, *Bamileke*, and *Bassa*) [7,8]. The patients were approached at the consultation services of the different health facilities, and then objectives of the study were explained before their inclusion upon signing an informed consent form. A questionnaire was administered to each participant to collect sociodemographic, anthropometric, and clinical information, while blood samples were collected to measure immune response parameters, respectively. The participants were recruited consecutively using a random sampling to reduce selection and information biases. A summary of the main activities conducted during the study is summarized in Fig 1.

### Eligibility criteria

We included all Cameroonian patients of both sexes, aged > 18 years old, settled permanently in Douala, and having signed an informed consent form. In contrast, we excluded from this study: i) foreigners, ii) patients who were not willing to participate, iii) those who refused to sign an informed consent form, iv) those admitted to intensive care units; v) COVID-19 vaccinated patients, and vi) those for whom blood collection was impossible.

## Sample size calculation

Participants were recruited consecutively using random sampling to limit selection and information biases. The sample size was determined using Lorentz's formula $n = [Z^2 \times p \times (1—p)]/d^2$, where $n$ = the required sample size, Z = statistics for the desired confidence interval (Z = 1.96 for 95% confidence level), d = accepted margin of error (d = 5%), and p = seroprevalence of SARS-CoV-2 antibodies in Africa (19.5%) [9]. The required sample size for this study was estimated as $n$ = 241 participants.

## Data collection

About three millilitres of whole blood were collected from each patient by venepuncture in properly labelled tubes. A structured pre-tested questionnaire was used to collect data of interest, which consisted of sociodemographic information (age, gender, educational level, occupation, and marital status), anthropometric parameters (weight, height, body mass index), and clinical information (clinical signs and symptoms, COVID-19 vaccination uptake, and presence of comorbidities such as diabetes and hypertension). Blood samples were collected from COVID-19 unvaccinated participants for determination of serum levels of immune response parameters (i.e., IgM, IgG, IFN-γ, IL-6, and CD4+), as per the objectives of the study.

## Determination of immune response parameters

Venous blood samples were used to determine levels of CD4+, immunoglobulins M and G (IgM and IgG), interferon gamma (IFN-γ) and interleukin 6 (IL-6). The blood level of CD4 + was determined using a Sysmex XF-1600 Flow Cytometer (https://www.sysmex-ap.com). Blood samples were centrifuged, and the resulting supernatant (serum) was used to determine levels of immunoglobulins, IFN-γ, and IL-6. All serum samples were tested for anti-SARS-CoV-2 virus IgG and IgM antibodies using two kits namely COVID-19 IgM ELISA (Calbiotech, USA, Reference CV461M) and COVID-19 IgG ELISA (Calbiotech, El Cajon, USA, Reference CV460G) which is an automated assay based on the enzyme-linked immunosorbent assay (ELISA) technique. Samples were considered positive for both IgG and IgM when the test values were greater than 1. Sensitivity and specificity values of the both assays were 93.75% and 97.3% for the COVID-19 IgM ELISA kit, 95.55% and 100% for the COVID-19 IgG ELISA kit. IFN-γ and IL-6 were measured by sandwich ELISA. Spectrophotometric measurements of IFN-γ and IL-6 were done at wavelength 450 nm. All experiments were performed in duplicate and in accordance with the manufacturer's instructions.

## Ethical statements

An ethical authorization was issued by the institutional review board of the University of Douala (N˚ 2945 CEI-UDo/12/2021/T), Littoral Health Regional Delegation (N˚ 0038/ AAR/MIN-SANTE/DRSPL/BCASS), and Douala Laquintinie Hospital (N˚ 08179/AR/MINSANTE/DHL) (S1 Fig). The study was explained to participants in the two official languages they understood best (French or English), and their questions were answered. Patients were informed about the objectives, advantages, and risks of the study, and then asked to sign a written informed consent form before their enrolment. Participants were informed that the study was strictly voluntary, and they were free to decline answering any question or totally withdraw if they so wished at any time.

## Statistical analysis

Data were presented as percentages with 95% confidence intervals (95%CI) for categorical variables, and mean ± standard deviation (SD) or standard error (SE) for continuous variables.

The percentages were compared using Fisher's exact and Pearson's independence chi-square ($\chi^2$) tests. Continuous variables were tested for normal distribution using the Kolmogorov-Smirnov test to decide whether parametric and non-parametric statistical tests were suitable for comparative analyses (S2 Fig). Parametric tests including one-way analysis of variance (ANOVA), post-hoc Duncan's test, unpaired samples Student t-test and Pearson correlation were used for variables following a Gaussian distribution. The non-parametric versions of these tests (i.e., Kruskal-Wallis test, Mann-Whitney test and Spearman correlation) were used for variables that failed to reach a Gaussian distribution. Given that there is a difference between the probability of being seropositive and the probability of a positive test, thus, it is crucial to take into account IgM/IgG kit sensitivity and specificity to adjust seroprevalence estimates. We therefore adjusted crude seroprevalence estimates for laboratory test kit error as proposed by Sempos and Tian [10], using the following formula:

Adjusted prevalence = (crude prevalence + specificity– 1)/(sensitivity + specificity—1), where the crude or observed prevalence is the proportion of the positive tests using the test kit, and sensitivity and specificity are their respective estimates.

Univariate and multivariate logit models were used to identify determinants of SARS–CoV–2 antibodies. The association between independent variables and SARS–CoV–2 antibodies was quantified by crude and adjusted odds ratios (cOR and aOR), 95%CIs, and level of statistical significance. The independent variables included in the logit model were health facility, gender, age, marital status, educational level, occupation, comorbidities, clinical signs and symptoms, blood group, COVID-19 vaccination uptake, history of past COVID-19, and history of recent infection). All variables used in the univariate analysis were used to build the multivariate logistic model [11].A two-tailed $p$-value $< 0.05$ was considered statistically significant. No imputation was performed for missing values. GraphPad version 5.03 for Windows (GraphPad PRISM, San Diego, Inc., California, USA), SPSS version 16 for Windows (SPSS IBM, Inc., Chicago, Illinois, USA), and StatView version 5.0 for Windows (SAS Institute, Chicago, Illinois, USA) software were used to perform statistical analyses. All analyses were performed as recommended in the SAMPL Guidelines [12].

## Results

### SARS-CoV-2 antibody response by COVID-19 status

Unvaccinated patients accounted for 81.4% of the patients (Fig 2A). The serum titres of IgM were significantly higher in unvaccinated patients compared to fully vaccinated patients (p = 0.03) (Fig 2B), but no significant difference was found for IgG (Fig 2C). Unvaccinated patients were analysed in the next sections as per the study objectives.

### Characteristics of the unvaccinated participants

Of the 342 participants, females and elderly people accounted for 47.7% and 13.2%, respectively. The mean age ± SD of the study population was 41.5 ± 13.9 years. Obesity was the main comorbidity (24.6%) found in participants. The bulk of participants (75.8%) were asymptomatic, even though a few individuals presented clinical symptoms represented mainly by cough (12.1%), severe fatigue (10.9%), and headache (9%) (Table 1).

### Overall crude and adjusted prevalence of SARS–CoV–2 antibodies among COVID-19 unvaccinated patients

A high proportion of patients were positive for both anti-SARS-CoV-2 antibodies, with an overall seroprevalence of 49.1% (n = 168, 95%CI 43.9–54.4%) for IgM, and 88.9% (n = 304,

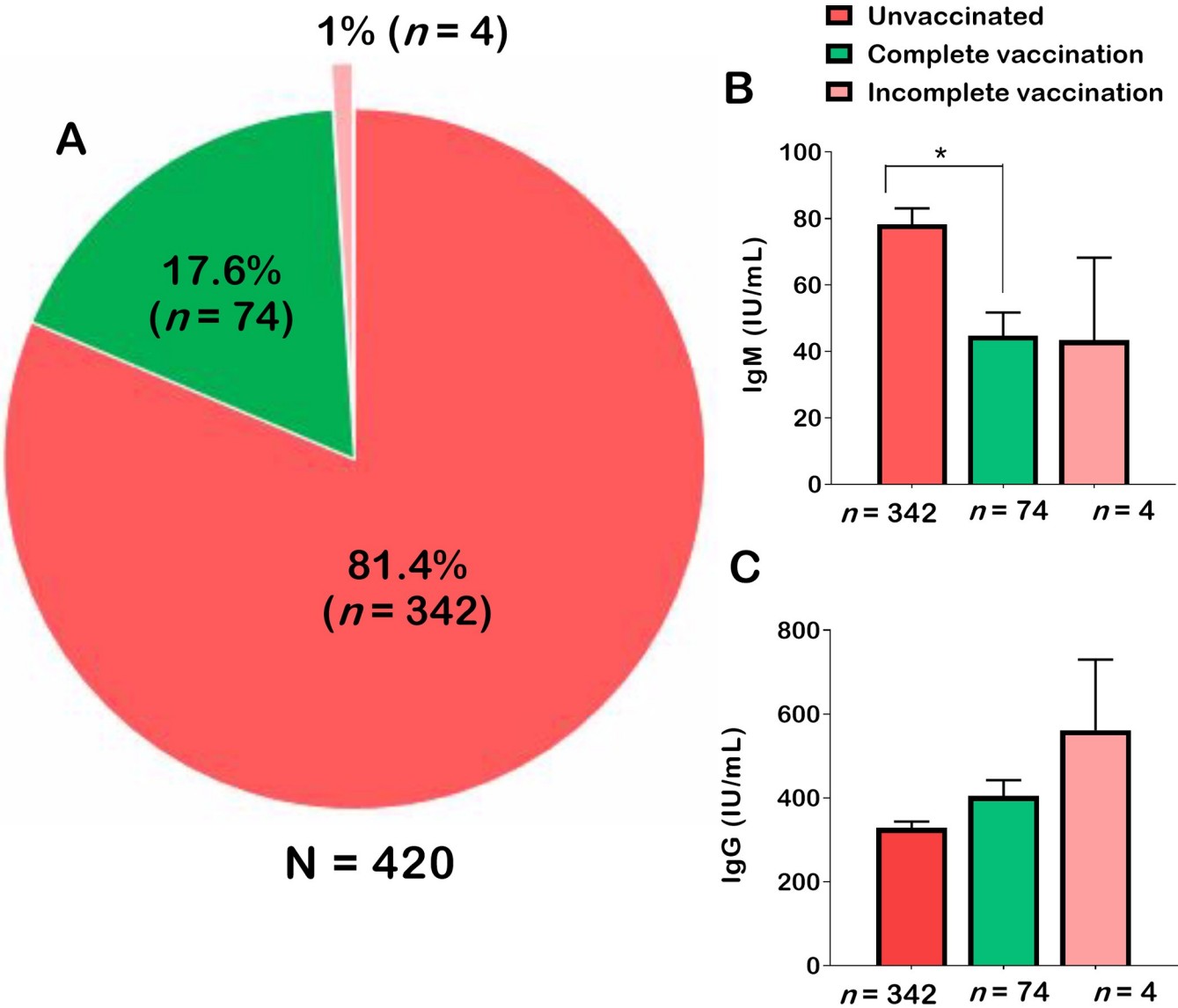

**Fig 2.** Proportion of unvaccinated patients (A), and serum levels of IgM (B) and IgG (C) by COVID-19 vaccination status. The one-way analysis of variance (ANOVA) and Duncan's post-hoc tests were used to make pairwise comparisons between groups. In Fig 2B and 2C, only statistically significant comparisons were showed on the graphs. *Statistically significant at $p$-value < 0.05.

95%CI 85.1–91.8%) for IgG. By combining the two antibodies, the seroprevalence was 96.8% ($n$ = 331, 95%CI 94.3–98.2%). After adjustment, the seroprevalence values were 51% for IgM and 93% for IgM. A statistically significant geographical variation in the prevalence of IgG, with the highest seroprevalence rate in patients from the Boko hospital (100%) and the lowest rate in patients from the Deido hospital (63.6%) (Fig 3 and S1 Table). No significant difference was found between the seroprevalence of IgM ($p$ = 0.51) or Ig G + IgM ($p$ = 0.22) after stratification by geographical area (Fig 3 and S1 Table). In contrast, a significant association was found between IgM-based SARS-CoV-2 seroprevalence and month of the study ($p$ = 0.04), with values of 54.5% (95%CI 34.7–67.1%) and 30% (95%CI 10.8–60.3%) in January and September, respectively (S1 Table). Finally, IgM- or IgG + IgM-based SARS-CoV-2 seroprevalences did not vary significantly over time (S1 Table).

**Table 1.** Details of the patients included in the study.

| Variables | *n* (%) |
|---|---|
| **Sociodemographic data (N = 342)** | |
| Females, *n* (%) | 163 (47.7%) |
| Age ≥ 60 years, *n* (%) | 45 (13.2%) |
| Mean age ± SD (years) | 41.5 ± 13.9 |
| Married, *n* (%) | 179 (52.3%) |
| University level, *n* (%) | 208 (60.8%) |
| **Comorbidities (N = 342)** | |
| Obesity, *n* (%) | 84 (24.6%) |
| Hypertension, *n* (%) | 37 (10.8%) |
| Diabetes, *n* (%) | 21 (6.1%) |
| Asthma, *n* (%) | 11 (3.2%) |
| Heart failure, *n* (%) | 8 (2.3%) |
| Human immunodeficiency infection, *n* (%) | 5 (1.5%) |
| Cancer, *n* (%) | 2 (0.6%) |
| Stroke, *n* (%) | 2 (0.6%) |
| Coronary heart disease, *n* (%) | 1 (0.3%) |
| Renal impairment, *n* (%) | 0 (0.0%) |
| **Clinical signs/symptoms (N = 342)** | |
| Asymptomatic, *n* (%) | 258 (75.8%) |
| Cough, *n* (%) | 49 (14.3%) |
| Severe fatigue, *n* (%) | 40 (11.7%) |
| Headache, *n* (%) | 35 (10.2%) |
| Fever, *n* (%) | 32 (9.4%) |
| Respiratory difficulties, *n* (%) | 27 (7.9%) |
| Sore throat, *n* (%) | 27 (7.9%) |
| Running nose, *n* (%) | 23 (6.7%) |
| Ageusia, *n* (%) | 22 (6.4%) |
| Anosmia, *n* (%) | 17 (4.9%) |
| Irritability/Confusion, *n* (%) | 9 (2.6%) |
| Loss of appetite, *n* (%) | 5 (1.5%) |
| Nausea, *n* (%) | 4 (1.2%) |
| Diarrhoea, *n* (%) | 4 (1.2%) |
| Vomiting, *n* (%) | 2 (0.6%) |

## Seroprevalence profile by clinical status among COVID-19 unvaccinated patients

The crude prevalence of IgG and/or IgM by clinical symptoms is depicted in Fig 4. Overall, the presence of IgM was more frequently seen among symptomatic patients. The trend was inverted regarding IgG antibodies, but there was no statistically significant difference for the concomitant presence of IgM + IgG. For instance, higher proportions of IgM+ patients were found in those with fever (71.9% *vs* 46.8%, *p* = 0.0008), ageusia (90.9% *vs* 46.3%, *p* < 0.0001), and anosmia (82.4% *vs* 47.4%, *p* = 0.005) (Fig 4 and S2 Table). Moreover, ageusia and anosmia were the COVID-19 symptoms that displayed the highest positive predictive values (90.9% and 82.4%) and specificity (98.9% and 98.3%) (S3 Table).

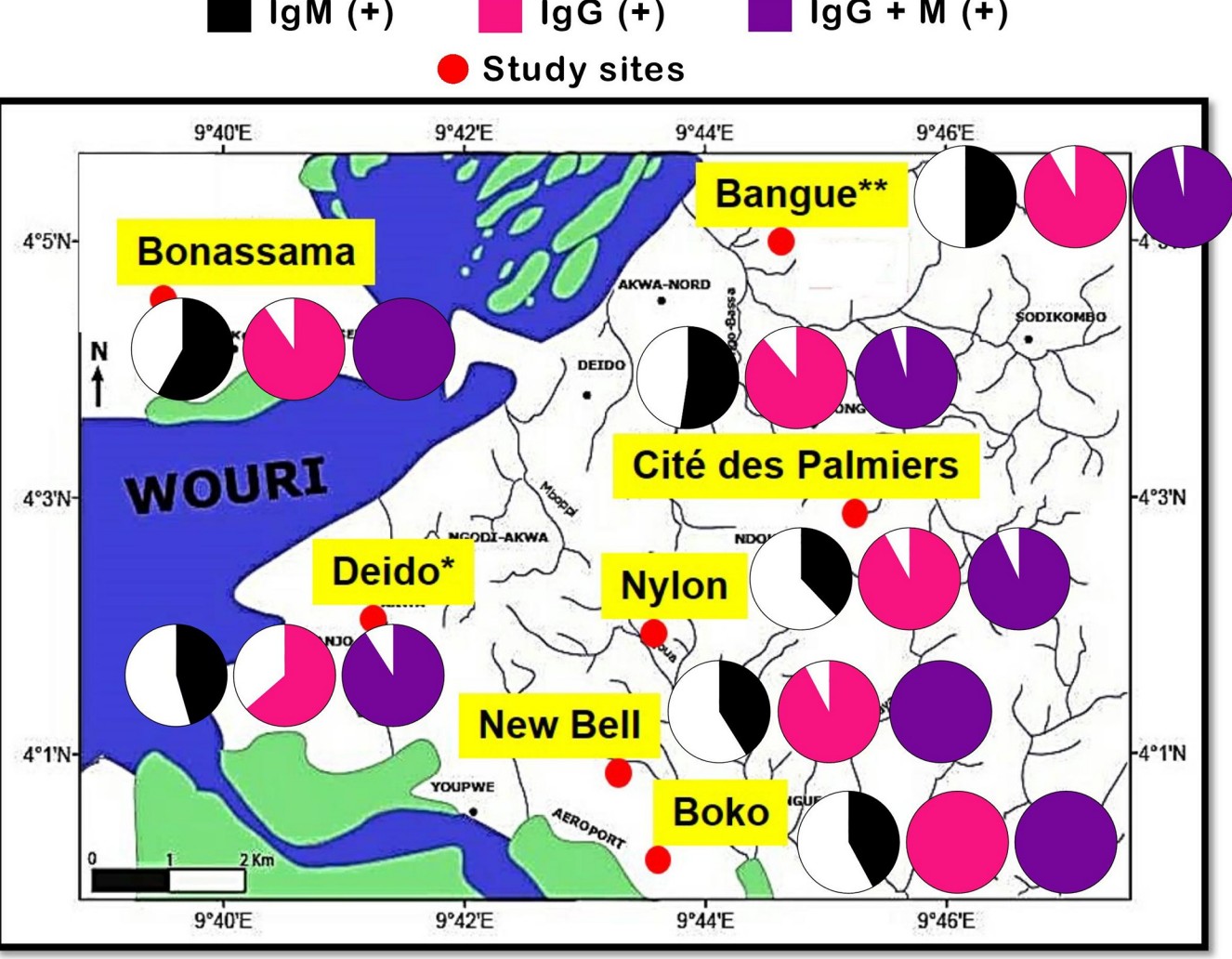

**Fig 3. Crude prevalence of SARS–CoV–2 antibodies by study sites.** The map was generated using AcgGIS v8.1 software (Esri, Redlands, CA, USA) and GraphPad version 5.03 for Windows (GraphPad PRISM, San Diego, Inc., California, USA). The pie charts depict the prevalence of patients positive for IgM (black section), IgG (pink section) and IgG + IgM (purple section). *For Deido district hospital, the sample collection site was located in Bonamoussadi neighbourhood. ** For Bangue district hospital, the sample collection site was located in Akwa neighbourhood.

## Seroprevalence by demographical information and comorbidities among COVID-19 unvaccinated patients

The variation of crude seroprevalence by demographical details and comorbidities is summarized in S4 Table. The prevalence of IgM was significantly higher in patients aged below 30 years (59.8%) and then decreased, with the lowest rates seen in patients 70 years (27.3%). The proportion of patients with IgG antibodies was higher in non-asthmatic patients as compared to their asthmatic counterparts (89.7% *vs* 63.6%, *p* = 0.02). No significant association was found between the presence of IgM and IgG antibodies, demographical characteristics, or comorbidities (S4 Table).

## Determinants of SARS-CoV-2 antibody response among COVID-19 unvaccinated patients

Based on univariate logistic regression analysis, a total of four factors associated with the presence of IgM antibodies were identified: age, marital status, frequent healthcare seeking, and

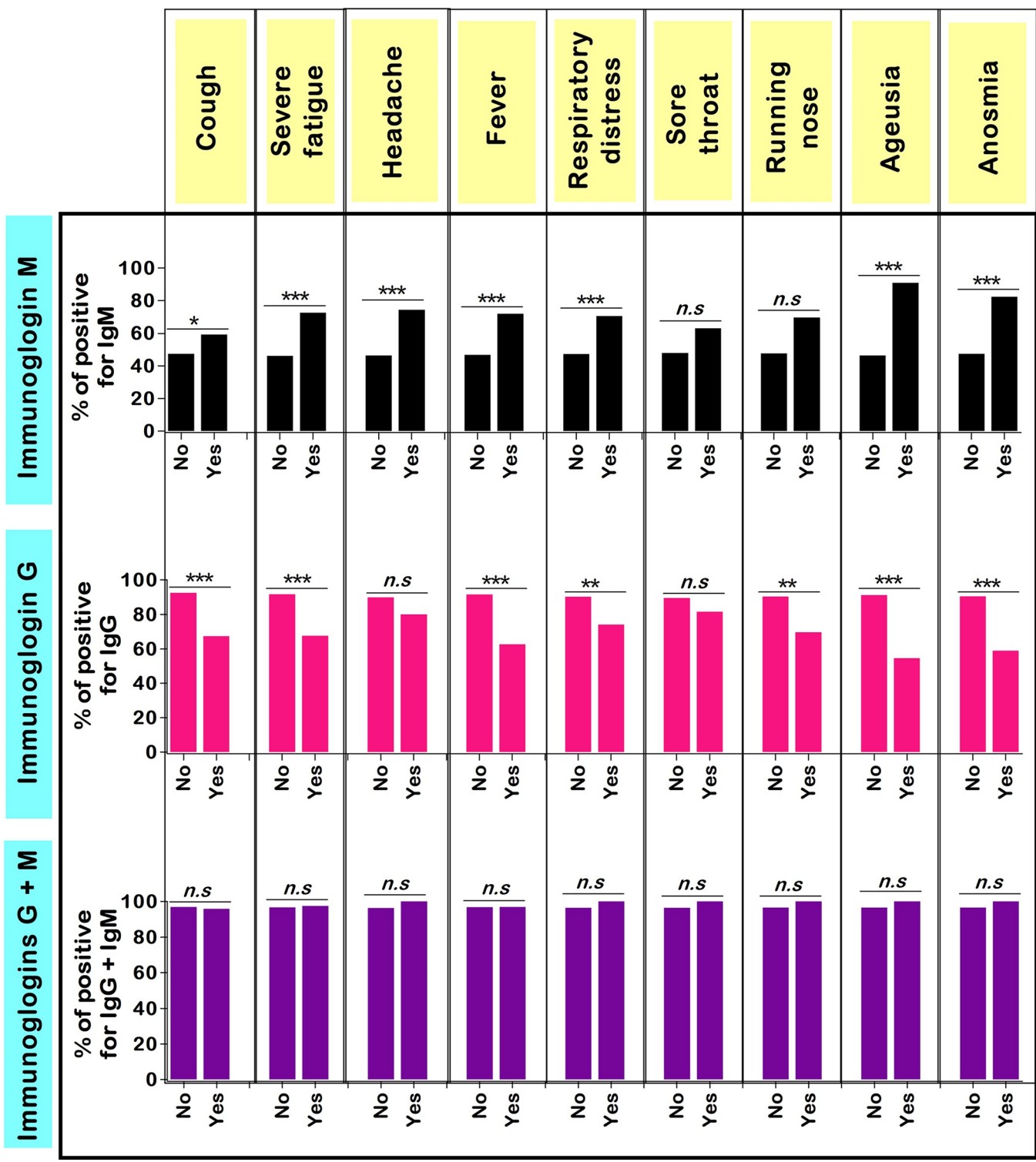

**Fig 4. Crude prevalence of SARS–CoV–2 antibodies by main clinical symptoms.** Ig: Immunoglobulin, n.s: Not significant, SARS–CoV–2: Severe acute respiratory syndrome coronavirus 2. Each bar represents the proportion of patients seropositive with respect to clinical symptoms. Only symptoms with occurrence > 15 were included in the analysis. Pearson's independence chi-square and Fisher's exact tests were used to compare percentages. *Statistically significant at *p-value < 0.05, **p-value < 0.01, and *** p-value < 0.0001.

occupation. The risk of the presence of IgM was reduced by 63% in patients aged 60–70 years (cOR = 0.37, 95%CI 0.16–0.84, *p* = 0.01) compared to those aged < 30 years. Similarly, the risk of IgM was reduced by 75% (cOR = 0.25, 95%CI 0.07–0.94, *p* = 0.04) in divorced or widowed patients, and by 51% (cOR = 0.49, 95%CI 0.25–0.98, *p* = 0.04) in patients working in the formal sector compared to singles and students, respectively. In contrast, the odds of being positive for IgM were nearly two-fold higher in frequently seeking care individuals (cOR = 1.71, 95%CI 1.09–2.66, *p* = 0.01) (Table 2). Regarding IgG, two factors were identified namely health facility and asthma. The risk of being positive for IgG was reduced in patients from the Deido hospital (cOR = 0.16, 95%CI 0.05–0.48, *p* = 0.002) compared to those recruited at the Bangue hospital. Likewise, the odds of being positive for IgG were reduced in asthmatic patients (cOR = 0.20, 95%CI 0.06–0.72, *p* = 0.01).

It was noted that the risk of being positive for IgM was increased but decreased for IgG in symptomatic patients, especially for those with fever, pain, respiratory difficulties, headache, severe fatigue, ageusia, or anosmia (Table 3). For instance, the IgM-related risk was increased in patients with fever (cOR = 2.91, 95%CI 1.30–6.49, *p* = 0.009), ageusia (cOR = 11.62, 95%CI 2.67–50.56, *p* = 0.0001), and anosmia (cOR = 5.18, 95%CI 1.46–18.38, *p* = 0.01). Conversely, the risk of being positive for IgG was reduced by 85% (cOR = 0.15, 95%CI 0.07–0.35, *p* < 0.0001), 88% (cOR = 0.12, 95%CI 0.05–0.29, *p* < 0.0001), and 85% (cOR = 0.15, 95%CI 0.05–0.42, *p* = 0.0003) in patients with fever, ageusia, and anosmia, respectively (Table 3).

## Predictors of anti-SARS-CoV-2 antibody response among unvaccinated patients

The predictors of the presence of IgM antibodies included diabetes, frequent healthcare seeking, and ageusia, while health facilities, older age, marital status, fever, and ageusia were predictors of the presence of IgG antibodies (Table 4). For instance, the risk of being positive for IgM was increased in patients frequently seeking health care (aOR = 1.97, 95%CI 1.05–3.68, *p* = 0.03) and those diagnosed with ageusia (aOR = 20.63, 95%CI 2.46–172.79, *p* = 0.005). Besides, the risk of IgG seropositivity was reduced by 85% in patients attending the Boko hospital (aOR = 0.15, 95%CI 0.03–0.72, *p* = 0.01), by 79% in married (aOR = 0.21, 95%CI 0.05–0.84, *p* = 0.02), by 92% in feverish patients (aOR = 0.08, 95%CI 0.01–0.64, *p* = 0.02), and by 92% those diagnosed with ageusia (aOR = 0.08, 95%CI 0.01–0.57, *p* = 0.01). In contrast, the odds of IgG seropositivity were increased in patients aged 40–50 years old (aOR = 8.78, 95%CI 1.54–50.01, *p* = 0.01) compared to those aged below 30 years old (Table 4).

## Variation of the SARS–CoV–2 immune response among unvaccinated patients

The variations in mean levels of CD4+ cells, IFN-γ, and IL-6 by seropositivity status are depicted in Fig 5. No significant variation was noted for CD4+ between IgM+ patients and IgM- patients (*p* = 0.11). In contrast, increased levels have been observed in IFN-γ (*p* = 0.01) and IL-6 (*p* = 0.04) in IgM+ patients compared to their IgM- counterparts. Regarding IgG status, the levels of CD4+ cells were significantly higher in IgG+ patients (*p* = 0.0048), but an inverted relation was found for IL-6 (*p* = 0.0001). The levels of IFN-γ were similar between IgG+ patients and IgG- patients (*p* = 0.87) (Fig 5).

The impact of patients' characteristics (i.e., sociodemographic, anthropometric, and clinical information) on the relation between seropositivity status, and CD4+ cells, IFN-γ, or IL-6 was analysed (Fig 6). For instance, we compared the mean values of CD4+ cells, IFN-γ, and IL-6 of IgM+ females with those of IgM- females, and the same comparisons were done for males. Overall, the relationship between CD4+ cells and IgM or IgG status was not influenced by

**Table 2.** Univariate logistic regression between SARS-CoV-2 antibodies, demographic information, clinical data and comorbidities among COVID-19 unvaccinated patients.

| Variables | Categories | IgM | | IgG | |
|---|---|---|---|---|---|
| | | cOR (95%CI) | *p* | cOR (95%CI) | *p* |
| **Sociodemographic characteristics** | | | | | |
| **Health facility** | Bangue | 1 | | 1 | |
| | Boko | 0.73 (0.27–1.95) | 0.53 | 0.71 (0.15–2.58) | 0.97 |
| | Bonassama | 1.38 (0.73–2.60) | 0.31 | 0.85 (0.29–2.51) | 0.76 |
| | Cité des Palmiers | 1.10 (0.59–2.05) | 0.76 | 0.73 (0.26–2.06) | 0.55 |
| | Deido | 0.83 (0.33–2.09) | 0.69 | 0.16 (0.05–0.48) | **0.001**\* |
| | New-Bell | 0.69 (0.33–1.46) | 0.34 | 1.09 (0.28–4.26) | 0.96 |
| | Nylon | 0.61 (0.26–1.41) | 0.25 | 0.44 (0.13–1.42) | 0.17 |
| **Age (years old)** | < 30 | 1 | | 1 | |
| | [30–40[ | 0.57 (0.31–1.04) | 0.06 | 1.39 (0.56–3.47) | 0.47 |
| | [40–50[ | 0.59 (0.31–1.15) | 0.06 | 1.83 (0.60–5.56) | 0.28 |
| | [50–60[ | 0.82 (0.41–1.66) | 0.58 | 1.43 (0.46–4.37) | 0.53 |
| | [60–70[ | 0.37 (0.16–0.84) | **0.01**\* | 0.90 (0.29–2.82) | 0.85 |
| | 70+ | 0.25 (0.06–1.02) | 0.05 | 0.70 (0.13–3.66) | 0.66 |
| **Gender** | Females | 1 | | 1 | |
| | Males | 0.76 (0.49–1.16) | 0.19 | 0.69 (0.35–1.37) | 0.28 |
| **Marital status** | Single | 1 | | 1 | |
| | Married | 0.72 (0.46–1.11) | 0.12 | 0.54 (0.26–1.11) | 0.09 |
| | Divorced/Widow | 0.25 (0.07–0.94) | **0.04**\* | 1.04 (0.12–8.73) | 0.96 |
| **Educational level** | None/Primary | 1 | | 1 | |
| | Secondary | 0.93 (0.37–2.32) | 0.88 | 0.83 (0.17–4.01) | 0.82 |
| | University | 0.98 (0.41–2.36) | 0.97 | 0.77 (0.17–3.49) | 0.73 |
| **Occupation** | Student | 1 | | 1 | |
| | Formal sector | 0.49 (0.25–0.98) | **0.04**\* | 0.63 (0.18–2.17) | 0.45 |
| | Informal sector | 0.59 (0.27–1.31) | 0.19 | 0.53 (0.13–2.07) | 0.35 |
| **Clinical characteristics** | | | | | |
| **Obesity** | No | 1 | | 1 | |
| | Yes | 0.72 (0.44–1.18) | 0.19 | 0.90 (0.42–1.94) | 0.78 |
| **Diabetes** | No | 1 | | 1 | |
| | Yes | 0.49 (0.19–1.26) | 0.14 | 0.73 (0.21–2.62) | 0.63 |
| **Hypertension** | No | 1 | | 1 | |
| | Yes | 0.87 (0.44–1.72) | 0.68 | 0.61 (0.23–1.56) | 0.29 |
| **Heart failure** | No | 1 | | 1 | |
| | Yes | 0.34 (0.07–1.69) | 0.19 | - | - |
| **HIV** | No | 1 | | 1 | |
| | Yes | 0.69 (0.11–4.16) | 0.68 | 0.49 (0.05–4.53) | 0.53 |
| **Asthma** | No | 1 | | 1 | |
| | Yes | 1.85 (0.53–6.43) | 0.33 | 0.20 (0.06–0.72) | **0.01**\* |
| **History of COVID-19** | No | 1 | | 1 | |
| | Yes | 0.79 (0.41–1.52) | 0.48 | 5.61 (0.75–42.01) | 0.09 |
| **Blood group** | A | 1 | | 1 | |
| | AB | 0.82 (0.28–2.45) | 0.72 | 1.04 (0.12–9.28) | 0.97 |
| | B | 0.63 (0.33–1.22) | 0.17 | 1.07 (0.29–3.98) | 0.91 |
| | O | 0.98 (0.58–1.63) | 0.93 | 0.41 (0.16–1.04) | 0.06 |
| **Frequently care seeking** | No | 1 | | 1 | |

*(Continued)*

**Table 2.** (Continued)

| | | IgM | | IgG | |
|---|---|---|---|---|---|
| **Variables** | **Categories** | **cOR (95%CI)** | **p** | **cOR (95%CI)** | **p** |
| | Yes | 1.71 (1.09–2.66) | **0.01**\* | 0.79 (0.39–1.59) | 0.51 |

95%CI: Confidence interval at 95%, COVID-19: Coronavirus disease 2019, cOR: Crude odds ratio, aOR: Adjusted odds ratio, HIV: Human immunodeficiency virus infection, Ig: Immunoglobulin, SARS–CoV–2: Severe acute respiratory syndrome coronavirus 2.

Univariate logistic regression analysis was performed to quantify the association between presence of anti-SARS-CoV-2 antibodies (IgG and IgM), demographical information, clinical data and comorbidities.

\*Statistically significant at *p*-value < 0.05.

**Table 3.** Univariate logistic regression analysis between presence of anti-SARS-CoV-2 antibodies and clinical signs/symptoms among COVID-19 unvaccinated patients.

| | | IgM | | IgG | |
|---|---|---|---|---|---|
| **Variables** | **Categories** | **cOR (95%CI)** | **p** | **cOR (95%CI)** | **p** |
| **Cough** | No | 1 | | 1 | |
| | Yes | 1.61 (0.87–2.97) | 0.13 | 0.17 (0.08–0.35) | **< 0.0001**\* |
| **Fever** | No | 1 | | 1 | |
| | Yes | 2.91 (1.30–6.49) | **0.009**\* | 0.15 (0.07–0.35) | **< 0.0001**\* |
| **Sore throat** | No | 1 | | 1 | |
| | Yes | 1.85 (0.82–4.16) | 0.14 | 0.51 (0.18–1.45) | 0.21 |
| **Pain** | No | 1 | | 1 | |
| | Yes | 2.86 (1.16–7.05) | **0.02**\* | 0.22 (0.09–0.56) | **0.001**\* |
| **Running nose** | No | 1 | | 1 | |
| | Yes | 2.51 (1.01–6.27) | **0.04**\* | 0.25 (0.09–0.64) | **0.004**\* |
| **Respiratory difficulties** | No | 1 | | 1 | |
| | Yes | 2.65 (1.13–6.22) | **0.02**\* | 0.31 (0.12–0.80) | **0.01**\* |
| **Diarrhoea** | No | 1 | | 1 | |
| | Yes | 1.04 (0.14–7.44) | 0.97 | 0.37 (0.04–3.64) | 0.39 |
| **Headache** | No | 1 | | 1 | |
| | Yes | 3.36 (1.52–7.40) | **0.002**\* | 0.45 (0.18–1.11) | 0.08 |
| **Severe fatigue** | No | 1 | | 1 | |
| | Yes | 3.09 (1.49–6.42) | **0.002**\* | 0.19 (0.09–0.41) | **< 0.0001**\* |
| **Irritability/Confusion** | No | 1 | | 1 | |
| | Yes | 2.11 (0.52–8.58) | 0.29 | 0.23 (0.06–0.98) | **0.04**\* |
| **Ageusia** | No | 1 | | 1 | |
| | Yes | 11.62 (2.67–50.56) | **0.001**\* | 0.12 (0.05–0.29) | **< 0.0001**\* |
| **Anosmia** | No | 1 | | 1 | |
| | Yes | 5.18 (1.46–18.38) | **0.01**\* | 0.15 (0.05–0.42) | **0.0003**\* |
| **Loss of appetite** | No | 1 | | 1 | |
| | Yes | - | - | 0.08 (0.01–0.48) | **0.005**\* |

95%CI: Confidence interval at 95%, cOR: Crude odds ratio, aOR: Adjusted odds ratio, Ig: Immunoglobulin, SARS–CoV–2: Severe acute respiratory syndrome coronavirus 2.

Univariate logistic regression analysis was performed to quantify the association between presence of anti-SARS-CoV-2 antibodies (IgG and IgM) and clinical signs/symptoms.

\*Statistically significant at *p*-value < 0.05.

**Table 4. Predictors of the presence of anti-SARS-CoV-2 antibodies among COVID-19 unvaccinated patients.**

| | | IgM | | IgG | |
|---|---|---|---|---|---|
| Variables | Categories | aOR (95%CI) | p | aOR (95%CI) | p |
| **Health facility** | Bangue | 1 | | 1 | |
| | Boko | 0.91 (0.31–2.63) | 0.85 | - | 0.97 |
| | Bonassama | 1.74 (0.80–3.78) | 0.16 | 0.52 (0.13–2.06) | 0.35 |
| | Cité des Palmiers | 1.91 (0.84–4.31) | 0.12 | 0.46 (0.10–2.11) | 0.31 |
| | Deido | 0.70 (0.21–2.36) | 0.56 | 0.15 (0.03–0.72) | **0.01**\* |
| | New-Bell | 0.89 (0.37–2.15) | 0.79 | 0.69 (0.14–3.48) | 0.65 |
| | Nylon | 0.68 (0.25–1.82) | 0.44 | 0.40 (0.09–1.91) | 0.25 |
| **Age (years old)** | < 30 | 1 | | 1 | |
| | [30–40[ | 0.84 (0.37–1.92) | 0.67 | 3.13 (0.73–13.43) | 0.12 |
| | [40–50[ | 0.89 (0.34–2.35) | 0.81 | 8.78 (1.54–50.01) | **0.01**\* |
| | [50–60[ | 1.45 (0.47–4.41) | 0.51 | 5.50 (0.73–41.45) | 0.09 |
| | [60–70[ | 0.51 (0.14–1.83) | 0.31 | 3.58 (0.47–27.42) | 0.21 |
| | 70+ | 0.71 (0.09–5.30) | 0.73 | 1.11 (0.06–21.91) | 0.94 |
| **Gender** | Females | 1 | | 1 | |
| | Males | 0.97 (0.55–1.71) | 0.90 | 0.63 (0.23–1.75) | 0.37 |
| **Marital status** | Single | 1 | | 1 | |
| | Married | 0.68 (0.35–1.33) | 0.26 | 0.21 (0.05–0.84) | **0.02**\* |
| | Divorced/Widow | 0.20 (0.03–1.19) | 0.07 | 3.96 (0.05–326.53) | 0.54 |
| **Occupation** | Student | 1 | | 1 | |
| | Formal sector | 0.64 (0.24–1.69) | 0.36 | 0.45 (0.06–3.21) | 0.42 |
| | Informal sector | 0.76 (0.24–2.39) | 0.63 | 0.20 (0.02–1.73) | 0.14 |
| **Obesity** | No | 1 | | 1 | |
| | Yes | 0.77 (0.40–1.46) | 0.42 | 0.72 (0.22–2.35) | 0.58 |
| **Diabetes** | No | 1 | | 1 | |
| | Yes | 0.23 (0.07–0.78) | **0.01**\* | 1.26 (0.21–7.46) | 0.79 |
| **Hypertension** | No | 1 | | 1 | |
| | Yes | 0.75 (0.29–1.94) | 0.54 | 0.66 (0.12–3.56) | 0.62 |
| **Heart failure** | No | 1 | | 1 | |
| | Yes | 0.14 (0.02–1.11) | 0.06 | - | - |
| **Asthma** | No | 1 | | 1 | |
| | Yes | 0.69 (0.12–3.96) | 0.67 | 0.22 (0.02–2.34) | 0.21 |
| **History of COVID-19** | No | 1 | | 1 | |
| | Yes | 1.20 (0.52–2.77) | 0.66 | 2.65 (0.28–25.10) | 0.39 |
| **Blood group** | A | 1 | | 1 | |
| | AB | 0.82 (0.20–3.30) | 0.78 | 2.04 (0.10–43.55) | 0.64 |
| | B | 0.85 (0.38–1.90) | 0.69 | 0.94 (0.18–4.97) | 0.94 |
| | O | 1.37 (0.71–2.63) | 0.34 | 0.37 (0.10–1.42) | 0.14 |
| **Frequently care seeking** | No | 1 | | 1 | |
| | Yes | 1.97 (1.05–3.68) | **0.03**\* | 2.61 (0.72–9.48) | 0.14 |
| **Cough** | No | 1 | | 1 | |
| | Yes | 0.45 (0.15–1.36) | 0.15 | 0.44 (0.08–2.31) | 0.33 |
| **Fever** | No | 1 | | 1 | |
| | Yes | 1.25 (0.32–4.85) | 0.74 | 0.08 (0.01–0.64) | **0.01**\* |
| **Sore throat** | No | 1 | | 1 | |
| | Yes | 1.78 (0.52–6.11) | 0.36 | 1.42 (0.25–8.06) | 0.69 |
| **Pain** | No | 1 | | 1 | |

*(Continued)*

**Table 4.** (Continued)

| Variables | Categories | IgM | | IgG | |
|---|---|---|---|---|---|
| | | aOR (95%CI) | *p* | aOR (95%CI) | *p* |
| | Yes | 1.15 (0.23–5.78) | 0.86 | 3.24 (0.30–34.50) | 0.33 |
| **Running nose** | No | 1 | | 1 | |
| | Yes | 1.23 (0.29–5.27) | 0.78 | 1.14 (0.11–11.78) | 0.91 |
| **Respiratory difficulties** | No | 1 | | 1 | |
| | Yes | 0.54 (0.12–2.46) | 0.42 | 1.31 (0.11–14.91) | 0.82 |
| **Diarrhoea** | No | 1 | | 1 | |
| | Yes | 0.12 (0.01–1.81) | 0.12 | 0.08 (0.00–6.19) | 0.25 |
| **Headache** | No | 1 | | 1 | |
| | Yes | 1.05 (0.27–4.05) | 0.93 | 3.67 (0.53–25.24) | 0.18 |
| **Severe fatigue** | No | 1 | | 1 | |
| | Yes | 2.04 (0.41–10.04) | 0.38 | 0.18 (0.02–1.98) | 0.16 |
| **Irritability/Confusion** | No | 1 | | 1 | |
| | Yes | 0.71 (0.09–5.45) | 0.74 | 1.94 (0.15–24.85) | 0.61 |
| **Ageusia** | No | 1 | | 1 | |
| | Yes | 20.63 (2.46–172.79) | **0.005**\* | 0.08 (0.01–0.57) | **0.01**\* |
| **Anosmia** | No | 1 | | 1 | |
| | Yes | 2.32 (0.22–25.03) | 0.48 | 16.02 (0.56–455.01) | 0.10 |

95%CI: Confidence interval at 95%, COVID-19: Coronavirus disease 2019, aOR: Adjusted odds ratio, aOR: Adjusted odds ratio, Ig: Immunoglobulin.

Multivariate logistic regression analysis was performed to identify the predictors of the presence of anti-SARS-CoV-2 antibodies (IgG and IgM).

\*Statistically significant at *p*-value < 0.05.

patients' characteristics, with the exception of symptomatology and diabetes. The mean levels of CD4+ were significantly reduced in IgM+ symptomatic patients compared to IgM- symptomatic patients (*p* = 0.01), but the difference was no longer significant between IgM+ asymptomatic patients and IgM- asymptomatic patients (*p* = 0.16). Likewise, the same pattern was noted upon stratification of the patients by diabetic status, with a significant difference in CD4+ cells in diabetic patients (*p* = 0.01) but no longer in non-diabetic patients (*p* = 0.20). CD4+ cells were significantly higher in IgM+ non-diabetic compared to IgM+ non-diabetic (*p* = 0.005), but not in diabetic patients (*p* = 0.56). The association between seropositivity status and IFN-γ or IL-6 was also modified by the patient's details (Fig 6).

## Discussion

The epidemiological situation of SARS–CoV–2 infection is still elusive in Cameroon, and this is mainly due to a lack of testing campaigns in community. Molecular methods are the gold standard for SARS–CoV–2 testing in populations. In developing countries, their implementation at a large scale is strongly hindered by the high cost of these methods, which are limited to a few research institutes and health facilities for research and small-scale diagnosis purposes. Antibody-based assays constitute an interesting alternative to molecular methods, especially for the rapid determination of the circulation of SARS–CoV–2. This study aimed at determining the seroprevalence and determinants of SARS–CoV–2 antibodies among patients attending major health facilities for the management of COVID–19 in Douala, Cameroon.

The overall crude seroprevalence of SARS–CoV–2 was 96.9%, which is consistent with those reported in other settings such as Chile (97.3%) [13]. In contrast, lower seroprevalence values were recently reported in Portugal (2.7–3.9%), Mozambique (3%), Bosnia and

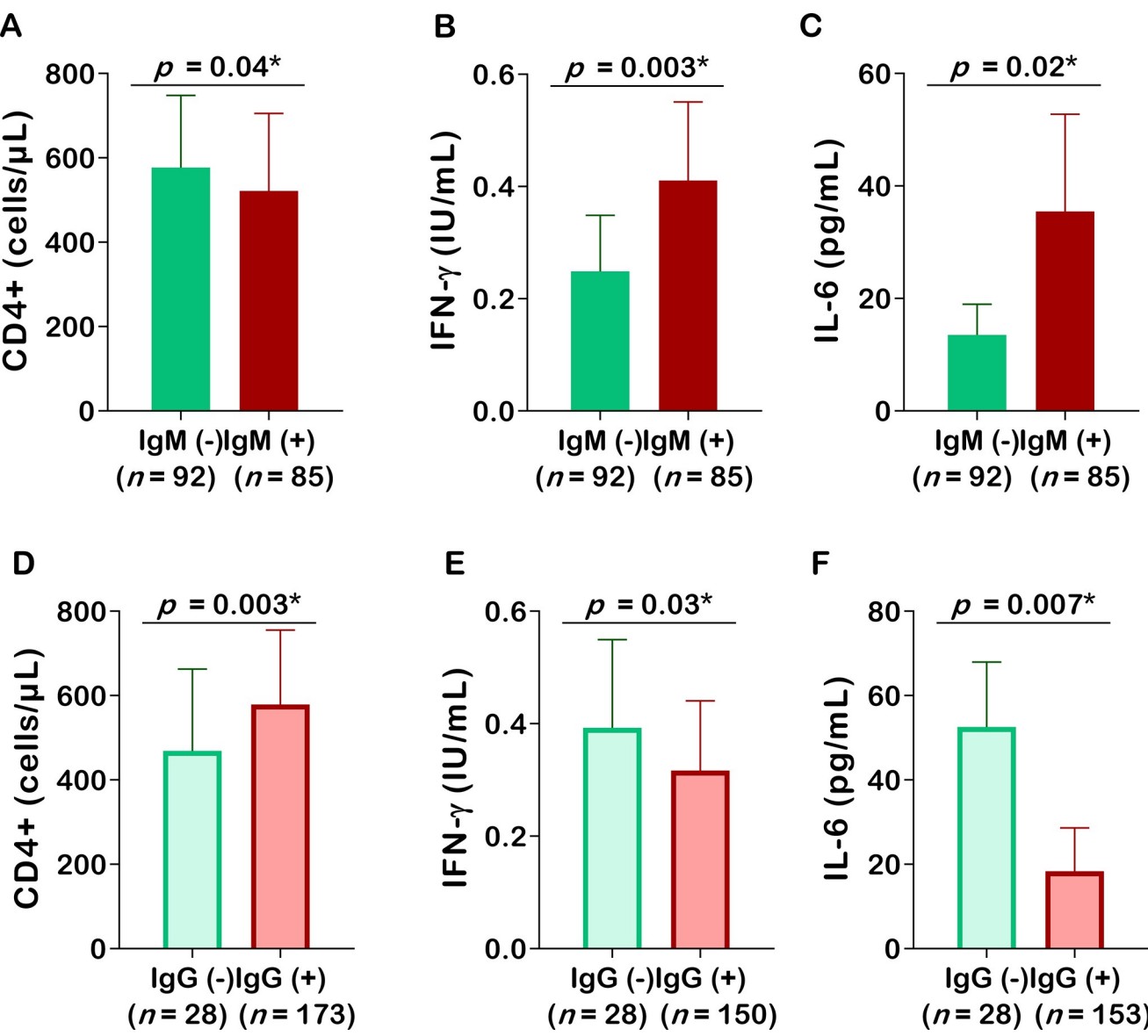

**Fig 5. Overall variation of CD4+ (A, D), IFN-γ (B, E) and IL-6 (C, F) with respect to presence of anti-SARS-CoV-2 IgM and IgG antibodies.** Ig: Immunoglobulin, IL-6: Interleukin 6, IFN-γ: Interferon gamma, CD: Cluster of differentiation. The parametric unpaired sample Student's t-test, and non-parametric Mann-Whitney test ere used to compare the groups. *Statistically significant at $p$-value < 0.05.

Herzegovina (3.77%), Japan (3.9%), Denmark (5.3%), Chile (7.2%), Spain (9.6–21.9%), Italia (11%), Sweden (11.8%), Iran (14%), Pakistan (16%), Iran (17.1%), Iraq (23.72%), Poland (42.7%), USA (5.6–57.7%), and Lebanon (58.9%) [4,14–30]. Differences in study period, SARS–CoV–2 variants, age groups, prevalence of risk factors and/or comorbidities (e.g., obesity, diabetes, hypertension), and COVID–19 vaccine coverage could explain discrepancies between seroprevalence estimates. For instance, peak periods of COVID-19 infection were different between the studies, and this is an important factor explaining the discrepancies between seroprevalence estimates. Also, we found higher seroprevalence against IgM compared to IgG. This is not in line with reports from Iraq, where authors found higher seroprevalence against IgG [4]. We noted that a high proportion of seropositive patients were

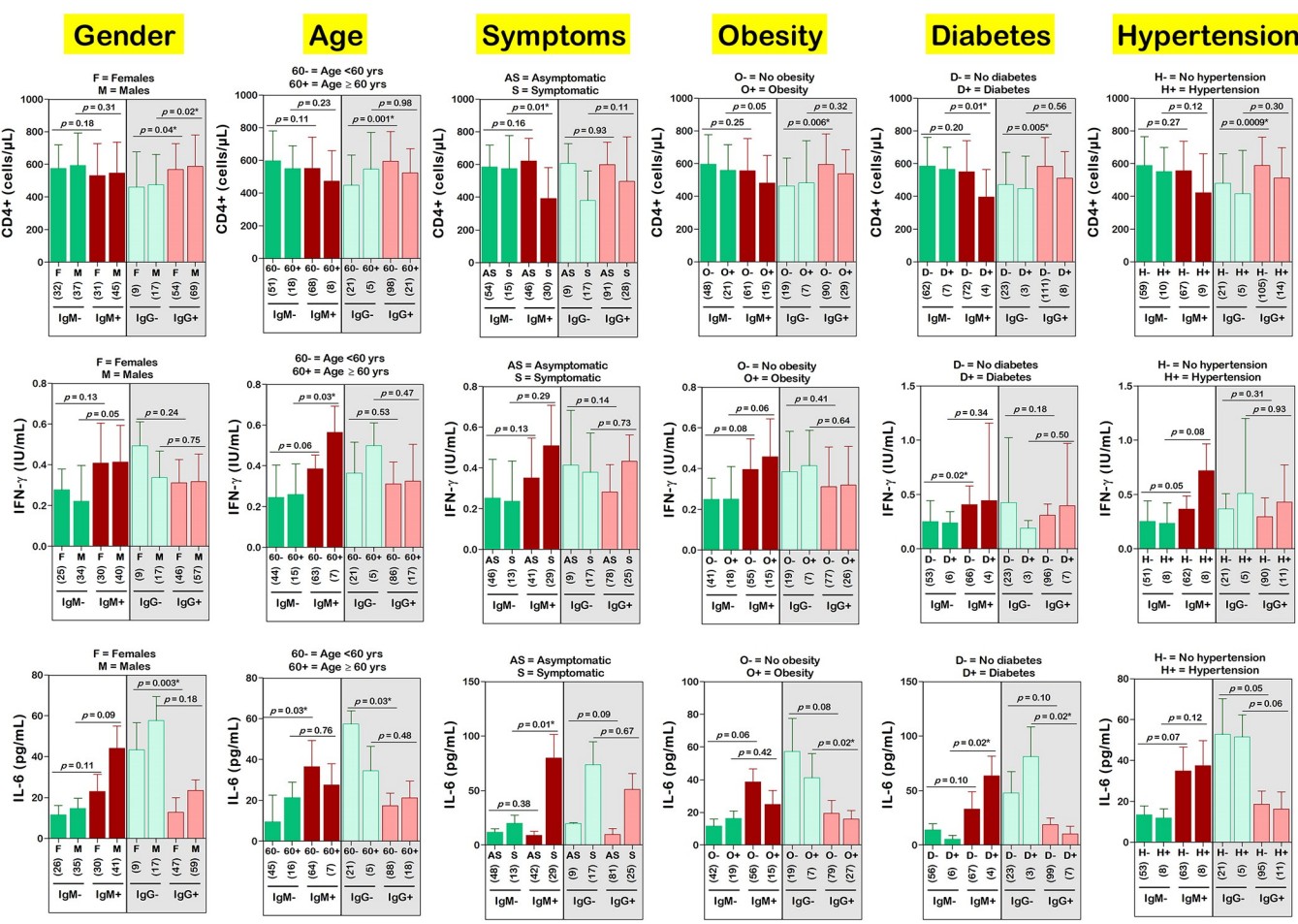

**Fig 6. Variation of CD4+, INF-γ and IL-6 by seropositivity status, demographic, clinical, and comorbidity information.** Ig: Immunoglobulin, SARS–CoV–2: Severe acute respiratory syndrome coronavirus 2, IL-6: Interleukin 6, IFN-γ: Interferon gamma, CD: Cluster of differentiation. The number of participants in each group are presented in round brackets. The non-parametric unpaired sample Student's t-test was used to compare the groups. *Statistically significant at $p$-value < 0.05.

asymptomatic, and this could indicate that patients could control the SARS-CoV-2 infection. A study found cross-reactive antibodies to SARS-CoV-2 were in circulation among HIV patients before the COVID-19 pandemics [31], thereby suggesting the presence of pre-existing anti-SARS-CoV-2- immunity that could contribute to attenuating disease severity.

The presence of IgM antibodies to SARS-CoV-2 is a good proxy for recent infection, as they are produced early in the antiviral humoral immune response. Nearly half of unvaccinated patients were IgM positive, thereby suggesting a high circulation of SARS-CoV-2 among the patients, and a need for adequate control measures, especially for those manifesting COVID-19-related symptoms. This is the first seroprevalence study to report SARS-CoV-2 risk in adults in Douala, the main populous and heterogeneous town of Cameroon. Before our study, IgM seroprevalence estimates were available from studies conducted among health care workers and the general population in the town of Yaoundé [32–34].

The crude and adjusted prevalence of IgG was 88.9% and 93% in this study. Such a high prevalence estimate was also reported in Cameroon by Ndongo Ateba *et al.* in Yaoundé (18.6% to 51.3%) [35], Mansuy *et al.* in pregnant women from Yaoundé (77%) [36], Njuwa Fai *et al.* in a community-based study from Yaoundé (24%) [34], Deutou Wondeu *et al.* among

university staff and students (71.3%) in Bandjoun, a kingdom in the West Region of Cameroon [37], Sandie *et al.* among blood donors from Douala and Yaoundé (66.3% to 98.4%) [38], and Diallo *et al.* among unvaccinated individuals (74.8%) after the Omicron wave [39]. Lower values were reported among dental teams in Germany (5.2%) [40] and young adults in Sweden (22.4%) [20]. The chances of detecting anti-SARS–CoV–2 IgG were decreased by 85% in patients attending the Deido district hospital compared to those attending the Bangue district hospital. This finding could be explained by geographical differences in the risk of infection, which have also been reported in previous studies [14]. In addition, the odds of IgG seropositivity were reduced in patients with fever or ageusia. IgG are markers of previous infection [41], and thus, it is likely that symptoms such as fever or ageusia could not be generally observed in patients with previous infection. Also, even though our findings suggest a certain clinical utility of fever or ageusia for diagnosis of previous SARS–CoV–2 infection, it should be noted that such symptoms are not pathognomonic to COVID-19/ SARS–CoV–2. Several authors pinpointed the limited clinical utility of symptom-driven surveillance or screening during COVID-19 pandemics [42].

Patients aged 60–70 years were less at risk of IgM seropositivity, which is consistent with earlier studies [17,25,30,39,43,44]. In Bosnia and Herzegovina, Prguda-Mujic and colleagues reported a reduced risk of positive anti-SARS CoV-2 Ig levels in patients aged over 50 years old [30]. Other studies reported a reduced risk of SARS-CoV-2 positivity in younger individuals [16,45]. This could be due to the fact that elderly individuals are more aware of risks of COVID–19, and thus preventive methods could be more effectively implemented in them [46]. This fact could also explain the reduced risk of anti-SARS CoV-2 IgG levels among asthmatic patients found in the present study.

In this study, clinical symptoms were significantly associated with a higher risk of IgM seropositivity, especially anosmia and ageusia. In addition, ageusia was the strongest predictor of IgM seropositivity among the patients. Several studies across COVID-19 burden varying settings also reported the clinical utility of COVID-19 evocating clinical symptoms, or more specifically, fever, anosmia, and ageusia [15,17,18,20,21,45,47]. Ferreira *et al.* recently reported that anosmia/dysgeusia was the most clinically discriminant symptoms in Portuguese municipal workers, with a PPV and specificity of 52.2% and 99.3%, respectively. Likewise, we found high estimates for anosmia (PPV = 82.4%, Sp = 98.3%) and for ageusia (PPV = 90.9%, Sp = 98.9%) in this study. The pathophysiological mechanism of anosmia and ageusia in COVID-19 is not yet fully understood, but studies have outlined that SARS-CoV-2 could elicit these olfactory and gustatory dysregulations either directly by infecting central nervous system and gustatory/olfactory epithelium cells, or indirectly through the production of cytokines such as tumour necrosis factor that may provoke the apoptosis of nervous cells [48,49].

In line with previous studies, the levels of IFN-γ and CD4+ cells were decreased in seropositive patients. Several studies outlined a delayed, decreased, or inhibited IFN-γ-mediated immune response in SARS–CoV–2 and other related coronaviruses (e.g., SARS–CoV–1). This dysregulation of the IFN-γ-mediated immune response is positively correlated with the severity of COVID–19 [50,51]. The pathophysiological mechanism through which SARS–CoV–2 provokes a dysregulation of the IFN-γ-mediated immune response included a production of antagonists that act by downregulating signalling pathways and/or inhibiting transcription factors [50,51]. Regarding CD4+ lymphocytes, they play a crucial role in the immune response against SARS–CoV–2 [52–54]. COVID–19 patients generally present a lymphopenia. Some authors suggested that the surveillance of lymphocyte subsets could be helpful for improved diagnosis and treatment of COVID–19 patients [55]. To be noted, the association between antibody seropositivity and CD4+, IFN-γ or IL-6 was modulated by patients' characteristics. Such modulating impact have been previously reported for immune response immune in

India [56], and haematological biomarkers of COVID-19 in Cameroon and India [57,58]. This is not surprising as these characteristics are well known to be predictors of SARS-CoV-2 infection, severity, or deaths [59–61].

The findings from the present study should be interpreted in light of its limitations. First, the study was conducted among inpatients from seven health facilities in the town of Douala, thereby limiting its generalisability at national level. Second, the study did not capture all environmental, behavioural, and socio-demographic characteristics of patients (e.g., size of household, socio-economic status) that could have impacted the risk of SARS-CoV-2 seropositivity [16,18,47,62–64]. Third, the small number of patients with comorbidities such as heart failure limited stratification analyses to identify possible confounding variables. Finally, the antigens in the ELISA kits used were designed to target the original SARS-CoV-2 lineage that emerged in China. Several variants of the virus, such as the Omicron or delta lineages, have been reported in Cameroon during pandemic waves [65–67], and thus, it is likely that a few infection cases were missed as these antigens may not optimally detect antibodies against SARS-CoV-2 variants and that the seroprevalence is underestimated.

## Conclusions

In this study, we aimed at determining seroprevalence, determinants of SARS-CoV-2 infection, and immunological alterations among unvaccinated patients living in Douala, Cameroon. The findings indicate a high circulation of the virus among participants, with several predictors of seropositivity including advanced older age, health facility, fever, frequent healthcare seeking, diabetes, marital status, and ageusia. In general, immune response effectors (CD4+, IFN-γ, and IL-6) analysed in the study were altered in seropositive individuals, with a confounding role of socio-demographic factors or comorbidities. Although the WHO recently declared the end of COVID-19 as a global health emergency, the findings of the present study indicate the need for continuous surveillance to adequately control the disease in Cameroon.

## Supporting information

**S1 Fig. Ethical clearance of the study: English translation and original clearance (French).**
(DOCX)

**S2 Fig. Testing SARS-CoV-2 immune response parameters (IgM, IgG, CD4+, IFN-γ and IL-6) and age for Gaussian distribution.**
(DOCX)

**S1 Table. Crude SARS-CoV-2 seroprevalence profile by months and health facilities.**
(DOCX)

**S2 Table. Crude SARS-CoV-2 seroprevalence profile by clinical symptoms.**
(DOCX)

**S3 Table. Clinical value of symptoms for prognostic of IgM seropositivity.**
(DOCX)

**S4 Table. Association between immune response and patients' characteristics among COVID-19 unvaccinated patients.**
(DOCX)

## Author Contributions

**Conceptualization:** Arlette Flore Moguem Soubgui, Martin Luther Koanga Mogtomo.

**Data curation:** Arlette Flore Moguem Soubgui, Wilfred Steve Ndeme Mboussi, Loick Pradel Kojom Foko, Elisée Libert Embolo Enyegue.

**Formal analysis:** Loick Pradel Kojom Foko, Martin Luther Koanga Mogtomo.

**Investigation:** Arlette Flore Moguem Soubgui, Wilfred Steve Ndeme Mboussi, Elisée Libert Embolo Enyegue.

**Methodology:** Arlette Flore Moguem Soubgui, Wilfred Steve Ndeme Mboussi, Loick Pradel Kojom Foko, Elisée Libert Embolo Enyegue.

**Project administration:** Martin Luther Koanga Mogtomo.

**Resources:** Martin Luther Koanga Mogtomo.

**Software:** Loick Pradel Kojom Foko.

**Supervision:** Martin Luther Koanga Mogtomo.

**Validation:** Loick Pradel Kojom Foko, Martin Luther Koanga Mogtomo.

**Visualization:** Arlette Flore Moguem Soubgui, Loick Pradel Kojom Foko, Martin Luther Koanga Mogtomo.

**Writing – original draft:** Arlette Flore Moguem Soubgui, Loick Pradel Kojom Foko.

**Writing – review & editing:** Loick Pradel Kojom Foko, Martin Luther Koanga Mogtomo.

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
