## [Decision Letter · Decision Letter 0]

20 Nov 2023

PGPH-D-23-01626

Serological surveillance reveals a high exposure to SARS-CoV-2 and altered immune response among COVID-19 unvaccinated Cameroonian individuals

Dear Dr. Kojom Foko,

Thank you for submitting your manuscript to PLOS Global Public Health. After careful consideration, we feel that it has merit but does not fully meet PLOS Global Public Health’s publication criteria as it currently stands. Therefore, we invite you to submit a revised version of the manuscript that addresses the points raised during the review process.

We look forward to receiving your revised manuscript.

Kind regards,

André Machado Siqueira, M.D., MSc, Ph.D

Academic Editor

Journal Requirements:

Additional Editor Comments (if provided):

Reviewers' comments:

Reviewer's Responses to Questions

**Comments to the Author**

1. Does this manuscript meet PLOS Global Public Health’s publication criteria? Is the manuscript technically sound, and do the data support the conclusions? The manuscript must describe methodologically and ethically rigorous research with conclusions that are appropriately drawn based on the data presented.

Reviewer #1: Partly

Reviewer #2: Yes

2. Has the statistical analysis been performed appropriately and rigorously?

Reviewer #1: I don't know

Reviewer #2: No

3. Have the authors made all data underlying the findings in their manuscript fully available (please refer to the Data Availability Statement at the start of the manuscript PDF file)?

Reviewer #1: Yes

Reviewer #2: No

4. Is the manuscript presented in an intelligible fashion and written in standard English?

Reviewer #1: Yes

Reviewer #2: Yes

5. Review Comments to the Author

Reviewer #1: This is an interesting study. Could you please correct or explain the following comments?

-Page8, Line 96: Informations missed:

How were the participants recruited;

What brought them to the selected health centers,

Why did you specifically choose these sites,

How did you approach the patients and suggest they take part in the study?

-Page 9, Line 121: Your study targeted only unvaccinated people. You need to better describe the approach, especially since the collection form (...Vaccination uptake...) mentions vaccination status and this is not one of the non-inclusion criteria.

-Page 10, line 132: Precision about the kit used for these tests missed

-Page 10, Line141: How did you deal with those who didn't understand either of these 2 languages? Especially since that wasn't one of your criteria for non-inclusion.

-Page 13, Line 193:Eliminate the dot at the end of table titles (check all)

-Page 25, Line355: To compare data with this country, you have to take into account the period, was it the same? Peak period of COVID infection? (Lines 355-357)

Reviewer #2: Thank you for the opportunity to review this interesting paper.

Am I correct in understanding that this study sampled only patients in inpatient settings? This limits the generalizability of the results presented here. On that note, I would ask why foreigners were excluded? And, how were subjects selected in the facility?

Re the sample size estimate: Sample size was chosen to ensure a precision of +/-5%. This is a relatively high error rate, especially when facility specific estimates are also given, which must have much larger error rates. For example, if there are 21 positive cases out of 22 subjects, the 95% CI extends from approximately 78.2% to 99.9%. For that reason, I'm not sure how useful it is to give facility specific estimates (and if that was the goal, the sample size should have been much higher in each facility).

Please report the sensitivity and specificity of the diagnostic tests for IgG and IgM. The analysis strategy should account for test accuracy (since the probability of a positive test is not exactly the same as the probability of being seropositive), as discussed e.g. in Sempos and Tian 2020 DOI: 10.1093/aje/kwaa174.

Confidence intervals should be given for the seroprevalence estimates.

The time frame for this study was quite long at 9 months. Can you make any statement about seroprevalence at the beginning of the study vs at the end?

Choice of variables in the multivariable models should be based on considerations related to e.g. confounding, not on univariable p-values. See Heinze and Dunkler 2017 DOI: 10.1111/tri.12895.

Figure 3 would be much improved if confidence intervals could be incorporated.

In Table S5, I'm not sure sensititivity, specificity etc are used correctly. Please check this.

The link to the data on Dryad did not work for me.

6. PLOS authors have the option to publish the peer review history of their article (what does this mean?). If published, this will include your full peer review and any attached files.

**Do you want your identity to be public for this peer review?** For information about this choice, including consent withdrawal, please see our Privacy Policy.

Reviewer #1: **Yes: **Tani SAGNA

Reviewer #2: **Yes: **Sarah Haile

---

## [Decision Letter · Decision Letter 1]

23 Jan 2024

Serological surveillance reveals a high exposure to SARS-CoV-2 and altered immune response among COVID-19 unvaccinated Cameroonian individuals

PGPH-D-23-01626R1

Dear Foko,

We are pleased to inform you that your manuscript 'Serological surveillance reveals a high exposure to SARS-CoV-2 and altered immune response among COVID-19 unvaccinated Cameroonian individuals' has been provisionally accepted for publication in PLOS Global Public Health.

Best regards,

André Machado Siqueira, M.D., MSc, Ph.D

Academic Editor

Reviewer Comments (if any, and for reference):

Reviewer's Responses to Questions

**Comments to the Author**

1. If the authors have adequately addressed your comments raised in a previous round of review and you feel that this manuscript is now acceptable for publication, you may indicate that here to bypass the “Comments to the Author” section, enter your conflict of interest statement in the “Confidential to Editor” section, and submit your "Accept" recommendation.

Reviewer #1: All comments have been addressed

Reviewer #2: All comments have been addressed

2. Does this manuscript meet PLOS Global Public Health’s publication criteria? Is the manuscript technically sound, and do the data support the conclusions? The manuscript must describe methodologically and ethically rigorous research with conclusions that are appropriately drawn based on the data presented.

Reviewer #1: Yes

Reviewer #2: Yes

3. Has the statistical analysis been performed appropriately and rigorously?

Reviewer #1: I don't know

Reviewer #2: Yes

4. Have the authors made all data underlying the findings in their manuscript fully available (please refer to the Data Availability Statement at the start of the manuscript PDF file)?

Reviewer #1: Yes

Reviewer #2: Yes

5. Is the manuscript presented in an intelligible fashion and written in standard English?

Reviewer #1: Yes

Reviewer #2: Yes

6. Review Comments to the Author

Reviewer #1: None

Reviewer #2: (No Response)

7. PLOS authors have the option to publish the peer review history of their article (what does this mean?). If published, this will include your full peer review and any attached files.

**Do you want your identity to be public for this peer review?** For information about this choice, including consent withdrawal, please see our Privacy Policy.

Reviewer #1: **Yes: **Tani SAGNA

Reviewer #2: No
